# The ER Unfolded Protein Response Effector, ATF6, Reduces Cardiac Fibrosis and Decreases Activation of Cardiac Fibroblasts

**DOI:** 10.3390/ijms21041373

**Published:** 2020-02-18

**Authors:** Winston T. Stauffer, Erik A. Blackwood, Khalid Azizi, Randal J. Kaufman, Christopher C. Glembotski

**Affiliations:** 1Department of Biology, San Diego State University Heart Institute, San Diego State University, San Diego, CA 92182, USA; wstauffe@gmail.com (W.T.S.); eblackwo@alumni.nd.edu (E.A.B.); azizik7@gmail.com (K.A.); 2Degenerative Diseases Program, Sanford Burnham Prebys Medical Discovery Institute, La Jolla, CA 92037, USA; rkaufman@sbpdiscovery.org; 3Department of Pharmacology, University of California, San Diego, La Jolla, CA 92161, USA

**Keywords:** ATF6, ER stress, endoplasmic reticulum, UPR, cardiac fibroblast, cardiac fibrosis, TGFβ, Smad

## Abstract

Activating transcription factor-6 α (ATF6) is one of the three main sensors and effectors of the endoplasmic reticulum (ER) stress response and, as such, it is critical for protecting the heart and other tissues from a variety of environmental insults and disease states. In the heart, ATF6 has been shown to protect cardiac myocytes. However, its roles in other cell types in the heart are unknown. Here we show that ATF6 decreases the activation of cardiac fibroblasts in response to the cytokine, transforming growth factor β (TGFβ), which can induce fibroblast trans-differentiation into a myofibroblast phenotype through signaling via the TGFβ–Smad pathway. ATF6 activation suppressed fibroblast contraction and the induction of α smooth muscle actin (αSMA). Conversely, fibroblasts were hyperactivated when ATF6 was silenced or deleted. ATF6 thus represents a novel inhibitor of the TGFβ–Smad axis of cardiac fibroblast activation.

## 1. Introduction

The adult heart consists of numerous cell types, including terminally differentiated cardiac myocytes, which are rigidly unable to respond to injury in a dynamic or proliferative sense. Accordingly, many therapeutic strategies focus on limiting damage rather than altering how heart cells change in the face of injury. However, while cardiac myocytes with a limited regenerative capacity contribute the bulk of the mass of the adult myocardium, a majority of the cells in the heart are proliferative and capable of differentiation or trans-differentiation during pathology [1].

The differentiation of progenitor cells, or the trans-differentiation of somatic cells into new cell types, puts increased demand on the capacity of the cells to synthesize and fold new proteins [2,3,4,5,6]. Of particular importance in responding to injury are secreted proteins, such as cytokines and extracellular matrix (ECM) proteins [7,8,9]. Many secreted proteins are translated on ribosomes on the endoplasmic reticulum (ER), which is the site of the translation and folding of at least 35% of new proteins, most of which move through the ER secretory pathway, destined for membrane insertion or secretion [10]. Conditions associated with myocardial injury also pose challenges to the folding environment of the ER by affecting redox status, calcium handling, and adenosine triphosphate (ATP) reduction [11,12,13]. These and other insults perturb ER proteostasis, the balance between the translation, folding, and degradation of ER proteins. The misfolded proteins that result from these challenges are detected by three ER-transmembrane proteins: protein kinase R-like ER kinase (PERK) [14], inositol-requiring protein-1 (IRE1) [15], and activating transcription factor-6 α (ATF6) [16,17]. Together, these sensors constitute the three major branches of the ER stress response, which is associated with a gene program that functions to preserve proteostasis in the ER, but then guides the cell toward apoptosis if protein misfolding is not resolved [11,18].

ER stress response pathways are important during the development and differentiation of certain cell types, especially professional secretory cells [2,4]. For example, in stimulated B lymphocytes, which differentiate into antibody-secreting plasma cells, activation of IRE1 and its downstream effector, x-box-binding protein 1 (XBP1), are necessary to accommodate the required expansion of ER membranes and increased protein trafficking [5]. ATF6 relatives in the old astrocyte specifically induced substance (OASIS) family are associated with differentiation in chondrocytes and osteoblasts by upregulating necessary differentiation factors [19]. Further, ATF6 itself was recently shown to drive induced pluripotent stem cells (iPSCs) toward mesenchymal lineages, while suppressing endo- and ectodermal lineages [6]. Lastly, ER stress signaling was shown to promote fibroblast-to-myofibroblast trans-differentiation in a manner similar to transforming growth factor β (TGFβ) signaling [20].

ER stress effectors, particularly ATF6, are critical for the mitigation of myocardial tissue damage and the preservation of heart function during heart disease [11,21]. ATF6 and ER stress additionally are shown to be important factors in the development of multiple tissue types [6,22,23]. The activation and differentiation of cardiac fibroblasts is a known feature of heart disease [7,8,9,24], however the role of ATF6 or the ER stress response in this process is not known. Here, we show that genetic markers of fibroblast activation, such as αSMA, are elevated in the heart following a myocardial infarction and that ATF6 deletion enhances this induction. We similarly found that loss of ATF6 causes the induction of genetic markers of myofibroblast activation, as well as enhancing TGFβ-mediated differentiation in primary isolated adult murine ventricular fibroblasts (AMVFs). Pharmacologically activating ATF6 in AMVFs had the opposite effect, decreasing myofibroblast markers. ATF6 activation in NIH 3T3 fibroblasts also inhibited the differentiation effect of TGFβ on fibroblast collagen contraction. These effects of ATF6 knockdown or deletion were found to be due to enhanced receptor-Smad phosphorylation, which was reversed upon pharmacological activation of ATF6. Inhibition of Smad signaling by ATF6 was correlated with the induction of various TGFβ/Smad pathway inhibitors, including Smad ubiquitination regulatory factor 1 (SMURF1), SMURF2, and prostate membrane protein androgen induced 1 (PMEPA1), and decreased levels of the TGFβ receptors, TGFβR1 and TGFβR2.

## 2. Results

### 2.1. Following Cardiac Injury, Mouse Hearts with ATF6 Deletion Increase Fibroblast Activation by Cardiac Injury

We previously showed that ATF6α, hereafter referred to as ATF6, is activated in mouse hearts subjected to a variety of insults, including ischemic and oxidative stress and in response to pressure overload [11,12,13,21]. In each of these instances, ATF6 was activated and was shown to reduce infarct size in response to ischemia/reperfusion and to promote compensatory hypertrophic cardiac growth during pressure overload, improving heart function. While these effects were partly due to the protective role of ATF6 in cardiac myocytes [11,12], the effects of ATF6 in non-myocytes in the heart were not known. Among the non-myocytes in the heart, fibroblasts play a major role during recovery from all these pathologies [7,8,9,24].

Ischemic injury stimulates fibroblast migration to the damaged area, after which they differentiate into myofibroblasts, which is characterized by induction of contractile proteins, such as αSMA, and extracellular matrix proteins, such as collagen 1a1 (Col1a1) [7,9,24,25]. Similarly, fibrosis is increased during pathological cardiac hypertrophy. Here, we found that, compared to control mouse hearts, αSMA and Col1a1 were significantly elevated in the hearts of ATF6 knockout (KO) mice subjected to myocardial infarction (Figure 1).

### 2.2. ATF6 Suppresses Genetic Markers of Fibroblast Activation in Response to TGFβ Treatment of Isolated Adult Murine Ventricular Fibroblasts (AMVFs)

Given the difference in fibrotic markers in ATF6 KO mouse hearts and the fact that fibroblasts are responsible for deposition of fibrotic material in the heart [9,25], we focused on investigating the role of ATF6 in cardiac fibroblasts. Accordingly, AMVFs were isolated (Appendix A) from WT and ATF6 KO mouse hearts, and then treated with 10 ng/mL TGFβ, a well-described cytokine that stimulates fibroblast activation [7,26,27], for 48 h. As expected, TGFβ increased *Acta2* and *Col1a1* mRNA in fibroblasts from WT mouse hearts, an effect that was significantly increased in fibroblasts from ATF6 KO mouse hearts (Figure 2A–C). Importantly, the effects of ATF6 deletion on TGFβ-mediated induction of fibroblast marker genes were recapitulated by siRNA-mediated knockdown (KD) of ATF6 in WT AMVFs (Figure 2D–F). In contrast, when ATF6 was activated by treating WT AMVFs with 10 μM compound 147, a small-molecule activator of ATF6 [28,29,30], for 48 h, there was a significant decrease of *Acta2* and *Col1a1* mRNA (Figure 2G–I). These results were also shown in NIH 3T3 cultures (Appendix A). Taken together, these results suggest ATF6 decreases expression of genetic markers of fibroblast activation.

### 2.3. ATF6 Activation Inhibits Fibroblast Contraction and Decreases αSMA Stress Fiber Formation in Response to TGFβ

As shown previously [7,8,9,24], when fibroblasts are activated, αSMA expression increases, and this leads to fibroblast contraction, a functional readout of fibroblast activation. This contraction can be measured when fibroblasts are embedded in a disk of polymerized collagen gel and subjected to stimuli causing fibroblast activation, where increases in contraction reduce the disk diameter [31]. To examine the function of ATF6 on fibroblast activation, TGFβ-mediated contraction of NIH 3T3 fibroblasts was assessed in the setting of ATF6 pharmacological activation. Strikingly, TGFβ-mediated contraction was decreased upon ATF6 activation with compound 147 (Figure 3A,B).

The inhibition of TGFβ-mediated fibroblast contraction by ATF6 activation implies that ATF6 exerts its effects at the level of αSMA contractile protein expression or structure. A major feature of fully activated myofibroblasts is the assembly of αSMA into contractile fibers, known as stress fibers, which are responsible for the contractile activity of the cell [7,24,27,32]. These fibers can be stained by fluorescent antibodies to αSMA and imaged as a final sign of full myofibroblast activation [24,33]. Accordingly, isolated AMVFs were treated with TGFβ with and without cotreatment with compound 147. As expected, compared to control, AMVFs treated with only TGFβ exhibited a significantly greater number of cells that had strongly expressed and fully assembled αSMA stress fibers. Intriguingly, cells treated with both TGFβ and 147 had far fewer stress-fiber-positive cells and were generally similar in appearance to control (Figure 3C,D), reflecting previously observed results on the mRNA level. These results were also shown in NIH 3T3 cultures (Appendix A). This demonstrates that the inhibition of TGFβ-mediated fibroblast contraction by ATF6 is at least in part due to the decreased formation of αSMA stress fibers in fibroblasts.

### 2.4. ATF6 Suppresses Smad2 Phosphorylation, a Measure of TGFβ-Mediated Fibroblast Activation, in Isolated AMVFs

To begin to dissect the level at which ATF6 affects TGFβ-mediated fibroblast activation, downstream events in the canonical TGFβ signaling pathway were interrogated. When TGFβ binds TGFβ receptors, the phosphorylation of receptor-Smad proteins 2 and 3 is increased, which leads to increased transcription of genes responsible for fibroblast activation [7]. Here, immunoblotting showed that, as expected, TGFβ increased the phosphorylation of Smad2 in WT AMVFs. Intriguingly, ATF6 knockdown (KD) (Figure 4A) slightly but significantly increased basal and TGFβ-mediated phosphorylation of Smad2 (Figure 4B), while activation of ATF6 with 147 (Figure 4C) reduced Smad2 phosphorylation (Figure 4D). These findings suggest that the inhibitory effect of ATF6 on fibroblast activation may be due to its ability to reduce TGFβ-mediated Smad2 signaling, though the potential for ATF6 to also act on other players in the TGFβ/Smad signaling pathway cannot be ruled out. To further confirm the involvement of ATF6 in the TGFβ/Smad signaling pathway in fibroblasts, AMVFs were treated with the TGFβ receptor inhibitor, SB431542. SB431542 reduced basal and TGFβ-mediated increases in *Acta2* and *Col1a1*, as well as completely blocking *Acta2* and *Col1a1* induction by ATF6 knockdown (Figure 5A–C). This further demonstrates that the fibroblast activation that takes place upon ATF6 loss-of-function is occurring because of a hyperactivation of the TGFβ/Smad pathway.

### 2.5. ATF6 Induces TGFβ/Smad Pathway Inhibitors and Suppresses Expression of TGFβ Receptors and Fibrosis-Related Genes

While the mechanisms by which ATF6 inhibits gene expression have not been explored, we considered it possible that, in fibroblasts, ATF6 might inhibit the expression of positive regulators of TGFβ/Smad signaling. Alternatively, ATF6 might increase the expression of negative regulators of the pathway. Accordingly, we examined the levels of several of these regulators in RNA isolated from AMVFs treated with 147. Among them, *Smurf1* [34], *Smurf2* [35], and *Pmepa1* [36,37], all negative regulators of TGFβ/Smad signaling, were the most significantly induced (Figure 6A). SMURF1 and 2 ubiquitylate R-Smads and target them for degradation [34,35]. PMEPA1 inhibits the TGFβ/Smad pathway by sequestering the R-Smads before they can be phosphorylated or interact with Smad4 [36,37]. *Pmepa1* is also heavily induced by TGFβ as a negative feedback loop. Importantly, *Smurf1* and *Pmepa1* transcript levels were significantly decreased when ATF6 was knocked down in AMVFs (Figure 6B).

To more broadly investigate the effect of ATF6 on fibrosis gene expressions, we performed PCR arrays interrogating the effects of ATF6 activation or knockdown in AMVFs. Consistent with the results above, we found that numerous pro-fibrosis genes were downregulated upon ATF6 activation with compound 147 (Figure 7A), and most of the same genes were upregulated upon ATF6 knockdown with siRNA (Figure 7B). Notable differentially regulated genes are summarized in Figure 7C. Several of the affected genes are directly involved in the TGFβ/Smad signaling pathway; when we perused the arrays for genes that were common to both treatments, we found that multiple isoforms of TGFβ itself were downregulated upon ATF6 activation and upregulated upon ATF6 knockdown, as were TGFβ receptors 1 and 2. RT-qPCR was used to validate these results, indicating that at least one way ATF6 affects fibroblast activation is by decreasing TGFβ and TGFβ receptor expression.

## 3. Discussion

This study shows that the ATF6 branch of the ER stress response has a role in regulating the activation of fibroblasts, including adult cardiac fibroblasts. This includes suppression of TGFβ–Smad2 signaling, part of the canonical pathway for fibroblast activation, which is summarized in Figure 8. The process of fibroblast-to-myofibroblast differentiation naturally involves the induction of numerous genes, including many whose protein products are translated on ER resident ribosomes and subsequently pass through all or part of the ER–Golgi secretory pathway [38]. This is especially true of the secretion of extracellular matrix proteins such as collagen, characteristic of the myofibroblast phenotype. As in the differentiation of other professional secretory cells such as plasma cells [39], the increased flux of newly translated proteins represents a challenge to ER proteostasis and can lead to activation of ER stress response pathways, including ATF6 [40]. ATF6, along with other ER stress pathways, was reported to be activated with TGFβ treatment [41]. Furthermore, amelioration of ER stress, through treatment with chemical chaperones 4-PBA or TUDCA, has been shown to reduce fibrosis and TGFβ signaling [42,43].

Several potential mechanisms are possible for how ATF6 activation regulates fibroblast activation. For example, one or more negative regulators of TGFβ–Smad2 could be non-canonical targets of ATF6. While ATF6 is primarily known for its role as a master regulator of the ER stress response and specifically of ER proteins involved in protein quality control, it is clear from recent publications that it is involved in many non-canonical pathways [11,12,44,45,46]. Jin et al. discovered that ATF6 governed an oxidative stress pathway by directly upregulating catalase, a well-known antioxidant targeted to the peroxisome [11]. More recently, Blackwood et al. [12] ound that ATF6 directly upregulates Ras homolog enriched in brain (Rheb), a cytosolic small GTPase that regulates mammalian target of rapamycin complex 1 (mTORC1) to induce cell growth. Furthermore, it was observed that the specific gene set upregulated by ATF6 varied depending on what the stimulus was [12]. Thus, general ER stressors such as tunicamycin (TM) activated ATF6 to induce the typical ER protein folding and degradation machinery but oxidative stressors caused ATF6 to also upregulate catalase, while cell growth signals caused an additional ATF6-mediated induction principally of Rheb. Another study by Tam et al. found that ATF6 was activated by the sphingolipids, dihydrosphingosine and dihydroceramide, which caused the upregulation of a previously unknown ATF6 gene set involved in the lipid biosynthesis pathway [45]. Thus, ATF6 induction of regulators of other non-ER stress pathways would not be unprecedented.

Though ATF6 is an activating transcription factor, it is also possible that, it directly downregulates a necessary component of TGFβ–Smad, such as TGFβ receptors or TGFβ itself. It was shown that numerous genes are perturbed in hearts with an activated form of ATF6 and that many of those are downregulated [44]. Moreover, Belmont et al. reported ATF6 is involved in the downregulation of many miRNAs, many of which did not target ER proteins and all of which lacked the consensus ATF6 binding site in their promoters, the ER stress response element (ERSE) [46]. Other studies found that ATF6 inhibits pathways by forming heterodimers with other transcription factors such as cAMP response element binding (CREB) [47] or sterol regulatory element binding protein 2 (SREBP2) [48] and altering their transcriptional activity. Thus, though ATF6 is an activating transcription factor, it can clearly participate in the inactivation of many genes, including some involved in fibroblast activation, as seen in Figure 8. Further study is needed to determine which of these genes, or a combination thereof, is necessary for ATF6 to confer this effect.

## 4. Materials and Methods

### 4.1. Laboratory Animal Use

The research reported here has been reviewed and approved, animal protocol 19-09-010G, approved 25 November 2019 by the San Diego State University Institutional Animal Care and Use Committee (IACUC) and conforms to the Guide for the Care and Use of Laboratory Animals published by the National Research Council.

### 4.2. Cardiac Non-Myocyte Isolation

Adult mouse ventricular fibroblasts (AMVFs) were isolated as previously described for adult ventricular myocytes [12] with the non-myocyte fraction cultured to grow a culture dominated by fibroblasts (Appendix A). Briefly, the ascending aorta of each mouse heart was cannulated, hung, and retrograde perfused at 3 mL/min for 4 min at 37 °C with heart medium (Joklik Modified Minimum Essential Medium (cat# M-0518, Sigma-Aldrich, St. Louis, MO, USA) supplemented with 10 mM HEPES, 30 mM taurine, 2 mM D-L-carnitine, 20 mM creatine, 5 mM inosine, 5 mM adenosine, and 10 mM butanedione monoxime (BDM), pH 7.36). Hearts were then digested by perfusing for 13 min with heart medium that contained type 2 collagenase (50–60 mg; ~320 U/mL, cat# LS004176, Worthington, Lakewood, NJ, USA) and 12.5 µM CaCl_2_. Beginning 5 min into the digestion, effluent from the heart was collected. The cannula was then removed, and the heart submerged in a dish with 2.5 mL of the accumulated effluent where the aorta and atria were excised. The remaining ventricles were then dissociated with forceps. The collagenase was neutralized with 2.5 mL of heart medium supplemented with 10% fetal bovine serum (FBS), and the final concentration of CaCl_2_ was adjusted to 12.5 µM. The now digested heart slurry was further dissociated into a cell suspension by gently and repeatedly pipetting up and down with a transfer pipette for 4 min. The cell suspension was then gravity filtered using a 100 µm mesh filter into a 50 mL conical tube. Following filtration, myocytes were sedimented by gravity into a pellet. After 6 min, the supernatant containing non-myocytes was carefully pulled off the myocyte pellet. The remaining myocytes in the pellet were resuspended and used for other experiments. The non-myocyte fraction was then spun down and resuspended in 10% FBS and plated on 12-well (0.5 mL per well) or 10 cm tissue culture dishes (10 mL per well). Following 16 h of incubation at 37 °C, the media was changed to remove non-viable cells and associated debris. The culture was then maintained for approximately 7 days, with further media changes every 2 days, until the cells were morphologically flat and spindle shaped and sufficiently dense to be used for experiments.

### 4.3. Immunoblotting

AMVFs were lysed in buffer comprising 50 mM Tris-HCl (pH 7.5), 150 mM NaCl, 0.1% SDS, 1% Triton X-100, protease inhibitor cocktail (Roche Diagnostics, Indianapolis, IN, USA) and phosphatase inhibitor cocktail (Roche Diagnostics). Lysates were subjected to centrifugation at 15,000× *g* for 15 min at 4 °C to pellet any cell debris, and the protein concentration was determined using the DC™ protein assay (Bio-Rad, Hercules, CA, USA). Samples usually comprising 10–30 µg of protein were mixed with Laemmli sample buffer including 2-mercaptoethanol, boiled for 5 min, then subjected to SDS-PAGE followed by transferring onto PVDF membrane for immunoblotting analysis. Antibodies were purchased that were raised against ATF6α (cat# 24169-1-AP, Proteintech, Rosemont, IL, USA, 1:1000), Smad 2/3 (cat# 3102, Cell Signaling, Danvers, MA, USA, 1:1000), phosphorylated Smad 2 (Ser456/467) (cat# 3108, Cell Signaling, Danvers, MA, USA, 1:1000), or GAPDH (cat# RDI-TRK5G4-6C5, Fitzgerald Industries International, Concord, MA, USA, 1:150 × 20,000).

### 4.4. Quantitative Real-Time PCR (qRT-PCR)

RNA was extracted from AMVFs or NIH 3T3 cultures or mouse heart tissue using Quick-RNA MiniPrep Kit, according to the manufacturer’s instructions (cat# R1055, Zymo Research, Irvine, CA, USA). cDNA was generated using Superscript III, according to the manufacturer’s instructions (cat# 18080-300, Life Technologies, Carlsbad, CA, USA). qRT-PCR was performed on an Applied Biosystems StepOnePlus™ Real-Time PCR System using the Maxima SYBR Green/ROX 2× qPCR Master mix (cat# K0223, Thermo Scientific, Waltham, MA, USA) and the mouse primers for *Acta2*, *Atf6*, *Col1a1*, *Pmepa1*, *Smurf1*, *Smurf2*, *Tcf21, Tnnt2*, and *Gapdh*. Sequences listed in Table 1 below:

### 4.5. PCR Arrays

PCR arrays were performed on cDNA generated using Qiagen RT^2^ First Strand Kit (cat# 330401, Qiagen, Germantown, MD, USA) from isolated AMVF RNA (described above) using RT^2^ Profiler PCR Arrays for Mouse Fibrosis (cat# PAMM-120Z, Qiagen, Germantown, MD, USA) according to the manufacturer’s instructions.

### 4.6. siRNA Transfection

Targeted gene knockdown was achieved with siRNA designed using Thermo Fisher Block-IT RNAi designer software (Thermo Fisher Scientific, Waltham, MA, USA). Transfection was achieved with HiPerFect transfection reagent (Qiagen, Germantown, MD, USA). Briefly, cell cultures were switched to 0.5%, antibiotic-free media that had been incubated at room temp for 15 min with HiPerFect and siRNA. Six microliters of 20 nM siRNA and 6 µL of HiPerFect were added per 1 mL of media and vortexed briefly prior to the 15-min incubation. siRNA was targeted to murine ATF6α (cat#10620312, Thermo Fisher Scientific, Waltham, MA, USA) and a non-targeting sequence (cat# 12935300, Thermo Fisher) was used as a control.

### 4.7. Collagen Gel Contraction Assay

NIH 3T3 cells were embedded in collagen gel disks, which were then suspended in culture media. Briefly, cells were resuspended in chilled serum-free culture media at a concentration of 150,000 cells/mL. For each gel, 400 μL of the cell suspension was combined with 200 µL of chilled 3 mg/mL Cultrex Rat Collagen I (R&D Systems, Minneapolis, MN, USA) and 12 μL of filtered 1 M NaOH as well as any experimental treatment. Five hundred microliters of this solution was quickly transferred to each well of a 24-well dish and incubated at 37 °C for 20 min. One milliliter of 10% FBS culture media, including any treatments, was then added to each well, and the polymerized gel disk was freed from the bottom of the disk using an autoclaved 200 μL pipette tip. The gels were verified to be freely floating before returning to the incubator. Contraction was then monitored at regular intervals up to 48 h.

### 4.8. Immunocytofluorescence

AMVFs or NIH 3T3s were plated and maintained as described above on four chamber glass slides (Falcon Brand, Corning, NY, USA). After cells had reached approximately 70% confluency, they were changed to media containing ±10 μM compound 147 and/or 10 ng/mL TGFβ for 48 h. After each treatment, AMVFs were washed with D-PBS, fixed for 15 min with 4% paraformaldehyde, and then permeabilized for 10 min with 0.5% Triton-X + 3 mM EDTA. Slides were blocked for 1 h with SuperBlock, and then incubated with a primary antibody to αSMA (MilliporeSigma cat #A2547, St. Louis, MO, USA, 1:500) for 16 h at 4 °C. Slides were incubated with Cy3 fluorophore-conjugated secondary antibodies for 1 h, followed by a Hoechst nuclear counter stain (1:1000) for 10 min.

### 4.9. Myocardial Infarction

Permanent occlusion myocardial infarction (MI) was performed in vivo by ligating the left anterior descending artery (LAD) as previously described [49]. Briefly, adult male mice were anesthetized using a 2% isofluorane/O_2_ mixture. Using aseptic technique, an incision was made to expose the trachea and the animal was intubated. Mice were then treated with buprenorphine (0.1 mg/kg IP) before an approximately 2 cm skin incision was made, lateral to the sternum and extending towards the axillary region. The left pectoralis muscle was retracted with an elastic retractor and the chest cavity penetrated with forceps in between the third and fourth ribs. Additional retractors were used to expose the anterior side of the left ventricle of the heart and the LAD. Using a 7–0 silk suture, the LAD was permanently ligated at a point just proximal to its downstream bifurcation and adjacent to the left atrium. Sham surgery mice were given the same procedure, except that the LAD was not ligated. The thoracic cavity and all skin incisions were then closed with surgical glue and the mice were transferred to individual cages to recover. Animals were again injected with buprenorphine (0.1 mg/kg IP) 12 h after surgery to aid in recovery and were continually monitored for signs of pain or distress thereafter. If such signs became apparent, additional doses of buprenorphine were administered. Seven days after surgery, animals were anesthetized, sacrificed, and the hearts removed. Hearts were dissected to separate atria and the right and left ventricle. The left ventricle was further separated into infarct, peri-infarct, and distal regions and then frozen in liquid nitrogen before transfer to a −80 °C freezer for storage.

### 4.10. WT and KO Mice

The ATF6 global knockout mice used in this study were 10-week-old male C57/Bl6 generated so that both ATF6α alleles had exons 5 and 6 globally deleted in all tissues and cell types, leading to a complete absence of any ATF6α protein but without affecting the other ATF6 isoform, ATF6β. ATF6α global knockout mice were a gift of Dr. Randall Kaufman and have been previously described [50]. ATF6-floxed mice were a gift from Gokhan S. Hotamisligil. Briefly, ATF6-floxed mice were generated with a targeting-construct flanking exons 8 and 9 of ATF6 with locus of X-over P1 (LoxP) sequences on a C57B/6J background, as previously described [51]. The plasmid encoding the human cardiac troponin T promoter driving Cre-recombinase was a gift from Dr. Oliver Müller [52]. Adeno-associated virus 9 (AAV9) preparation and injection were carried out as previously described [11,50]. Eight-week-old ATF6-floxed mice were injected via the lateral tail vein with 100 µL of AAV9-control or AAV9-cTnT-Cre containing 1 × 10^11^ viral particles and housed for 2 weeks before either sacrifice or experimental initiation, as previously described.

### 4.11. TGFβ, SB431542, and Compound 147 Treatment

All treatments applied to cell cultures were added in 10% FBS culture medium. Porcine, platelet-derived TGFβ1 (cat# 101-B1) and TGFβR1 inhibitor compound SB431542 (cat# 1614) were purchased from R&D Systems (Minneapolis, MN, USA). ATF6α activator compound 147 was a gift of Dr. Luke Wiseman and Dr. Jeffrey Kelly and has been previously described [28].

### 4.12. Statistics

All error bars shown are ±SEM and statistical treatments are Student’s *t* test when comparing two values or one-way analysis of variance (ANOVA) with Newman–Keuls post hoc analysis when comparing more than two values.

## 5. Conclusions

Activated ATF6 antagonizes the activation of cardiac fibroblasts in response to TGFβ treatment. This is evident as hyper-induction of the transcripts of genetic markers of fibroblast activation in the absence of ATF6 and their inhibition when ATF6 is activated. The assembly of characteristic myofibroblast αSMA stress fibers is also inhibited by ATF6, in agreement with the observation that fibroblasts with activated ATF6 are less able to contract. This phenomenon was made manifest when mouse hearts without ATF6 subjected to myocardial infarction showed greater induction of fibroblast activation markers in comparison to their WT counterparts. The identification of ATF6 as a novel and important component in the cardiac fibrosis pathway may be of great interest as a future therapeutic target.

## Figures and Tables

**Figure 1 ijms-21-01373-f001:**
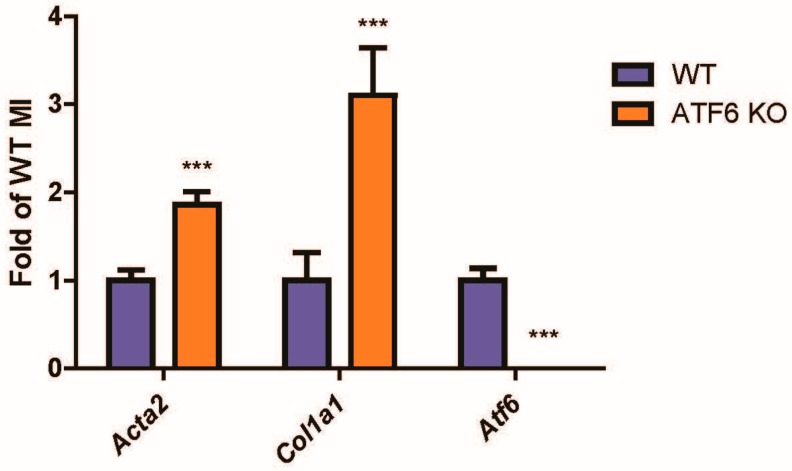
Markers of fibroblast activation in activating transcription factor-6 α (ATF6) knockout (KO) hearts of mice subjected to myocardial infarction (MI) surgery. Wild-type (WT) and ATF6 KO mice were subjected to permanent occlusion MI for one week. mRNA from infarcted regions in the hearts was then examined by qRT-PCR for genes indicative of cardiac fibroblast (CF) activation, α smooth muscle actin (αSMA) (*Acta2*) and collagen (*Col1a1*), as well as ATF6 (*Atf6*). *** *p* ≤ 0.001 significant difference from WT by Student’s t-test.

**Figure 2 ijms-21-01373-f002:**
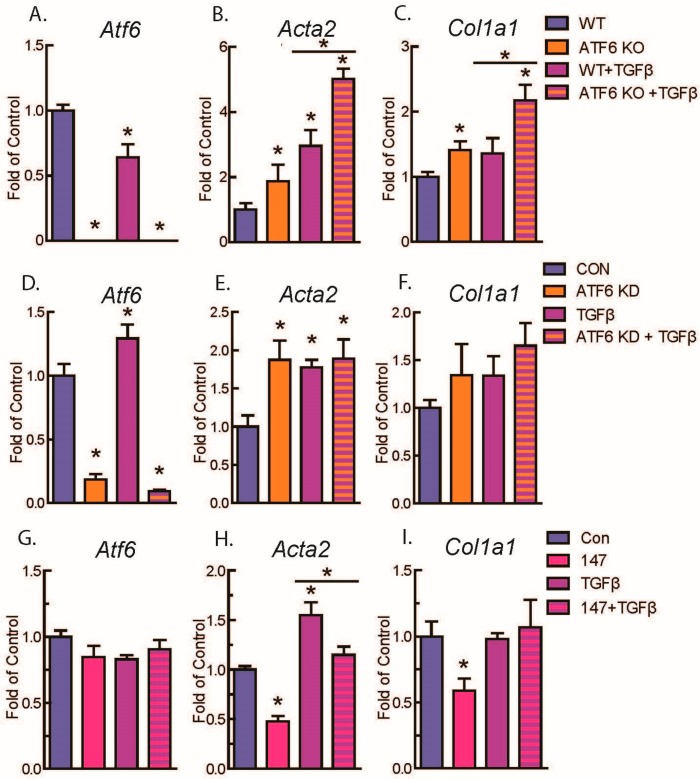
qRT-PCR of adult murine ventricular fibroblasts (AMVFs) with ATF6 gain- or loss-of-function. (**A**–**C**) WT and ATF6 KO AMVFs were treated with ±10 ng/mL transforming growth factor β (TGFβ) for 48 h, then analyzed by qRT-PCR for *Atf6*, *Acta2*, and *Col1a1*. (**D**–**F**) AMVFs from WT mouse hearts were treated with ±siRNA targeted to murine ATF6. Control (CON) and siRNA-treated cultures (ATF6 KD) were treated with ±10 ng/mL TGFβ, then analyzed by qRT-PCR for *Atf6*, *Acta2*, and *Col1a1*. (**G**–**I**) AMVFs from WT mouse hearts were treated with ±10 μM compound 147, a pharmacological activator of ATF6. Control (CON) and 147-treated cultures (147) were co-treated with ±10 ng/mL TGFβ for 48 h, then analyzed by qRT-PCR for *Atf6*, *Acta*, and *Col1a1*. * *p* ≤ 0.05 by one-way ANOVA. * Indicates significant difference between a condition and control according to Newman–Keuls post-test unless there is a line over two bars, which indicates those two bars are being compared as part of the same post-test.

**Figure 3 ijms-21-01373-f003:**
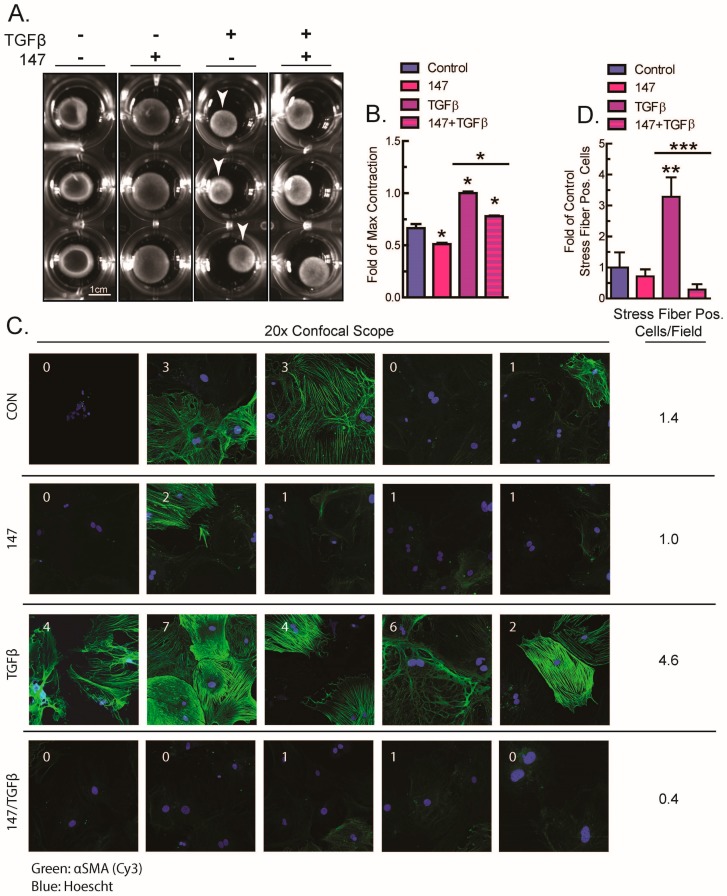
Effects of activating ATF6 on fibroblast contraction and stress fiber formation. (**A**,**B**) NIH 3T3 fibroblasts embedded in collagen gel disks were treated with ±10 μM compound 147, and then analyzed for contraction with ±10 ng/mL TGFβ after 48 h. Shown in (**A**) are examples of each culture and shown in (**B**) is the quantification of n = 3 cultures of each type, normalized to the cultures with maximum contraction (white arrows). (**C**,**D**) AMVFs from WT mice were treated with ±10 μM compound 147 and ±10 ng/mL TGFβ for 48 h, then analyzed by actin staining for stress fiber formation. All images in (**C**) were taken with a 20× objective on a confocal microscope. In (C) the number in each field represents the number of cells that were stress-fiber positive in that field. The number to the right is the average number of stress-positive cells per field. The number of stress-positive cells per field in (**C**) is quantified in (**D**), across n = 5 fields. * *p* ≤ 0.05 and ** *p* ≤ 0.01 by one-way ANOVA. *, ** Indicate significant difference between a condition and control according to Newman–Keuls post-test unless there is a line over two bars, which indicates those two bars are being compared as part of the same post-test.

**Figure 4 ijms-21-01373-f004:**
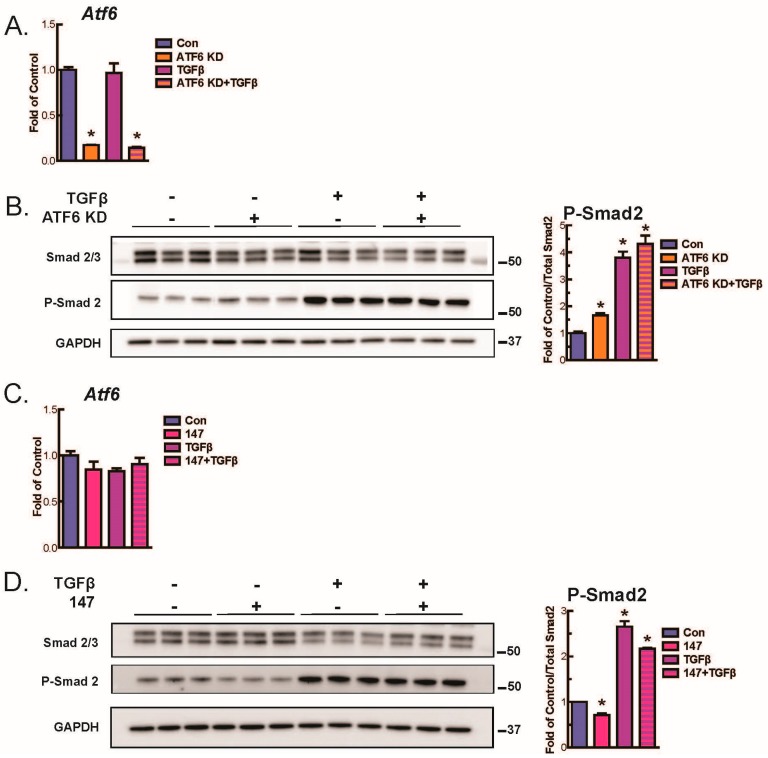
Immunoblots investigating activation of canonical TGFβ signaling pathways. (**A**) AMVFs were treated with ±siRNA to ATF6, ±10 ng/mL TGFβ for 48 h. *Atf6* knockdown is quantified via qPCR. (**B**) Immunoblotting for total Smad 2/3, P-Smad 2, or glyceraldehyde 3-phosphate dehydrogenase (GAPDH), with quantification of P-Smad 2 shown at right. (**C**) AMVFs were treated with ±10 μM compound 147, ±10 ng/mL TGFβ for 48 h. *Atf6* levels were quantified via qPCR. (**D**) Immunoblotting for total Smad 2/3, P-Smad 2, or GAPDH, with quantification of P-Smad 2 shown at right. * *p* ≤ 0.05 by ANOVA. * Indicates significant difference between a condition and control according to Newman–Keuls post-test.

**Figure 5 ijms-21-01373-f005:**
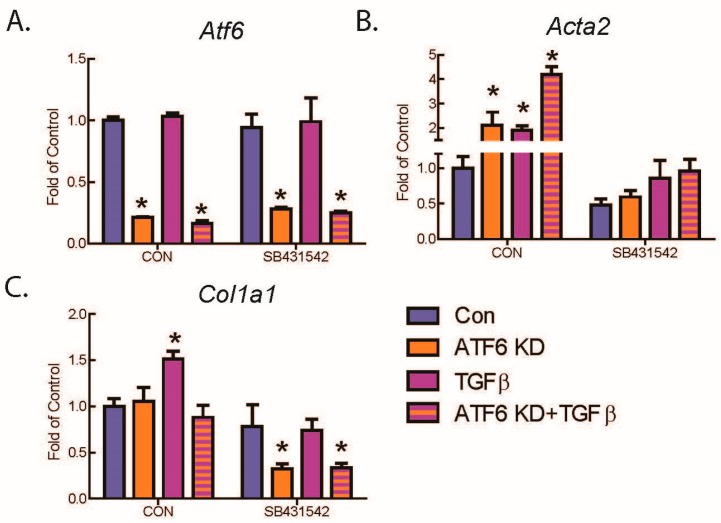
Effect of inhibiting the TGFβRI on ATF6 and TGFβ-mediated increases in fibroblast markers. AMVFs were treated with ±siRNA to ATF6, ±10 ng/mL TGFβ, ±10 μM of the TGFβRI inhibitor SB431542, and then analyzed by qRT-PCR for (**A**) *Atf6*, (**B**) *Acta2*, and (**C**) *Col1a1*. All treatments were for 48 h. * *p* ≤ 0.05 by one-way ANOVA. * Indicates significant difference between a condition and control according to Newman–Keuls post-test.

**Figure 6 ijms-21-01373-f006:**
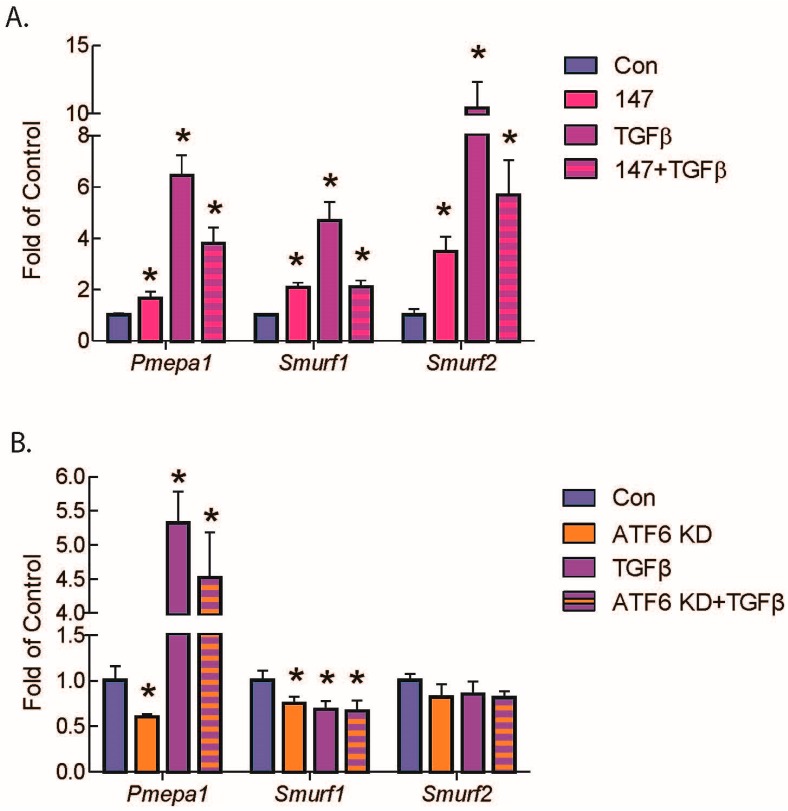
qRT-PCR of AMVFs treated with compound 147 or ATF6 knockdown and/or TGFβ. AMVFs were treated with (**A**) ±10 μM compound 147, ±10 ng/mL TGFβ, or (**B**) ±siRNA to ATF6, ±10 ng/mL TGFβ, then analyzed by qRT-PCR for *Pmepa1, Smurf1,* and *Smurf2*, as shown. All treatments were for 48 h. * *p* ≤ 0.05 by one-way ANOVA within each gene group. * Indicates significant difference between a condition and control according to Newman–Keuls post-test.

**Figure 7 ijms-21-01373-f007:**
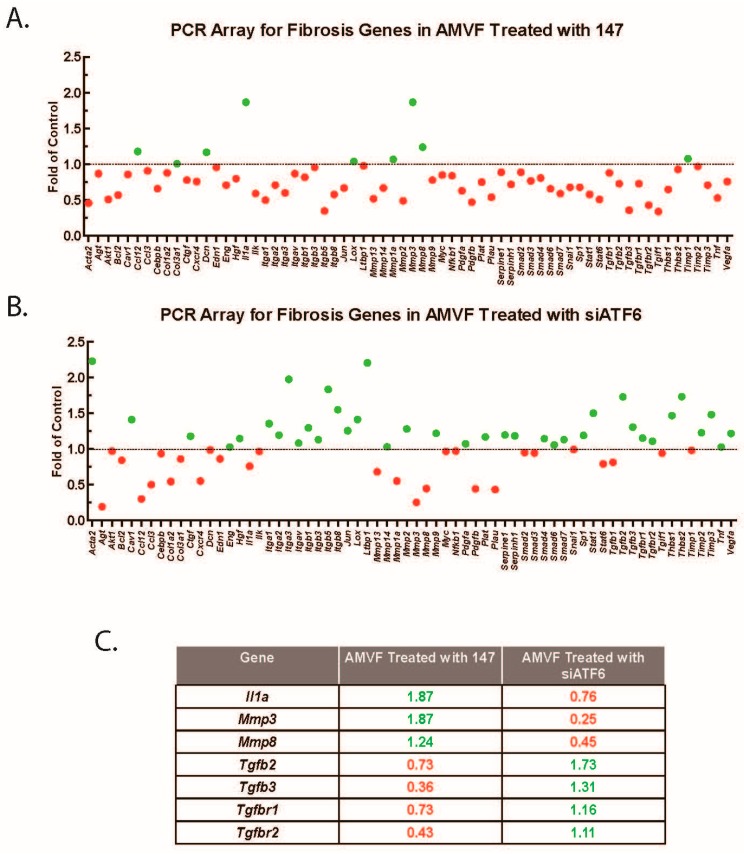
PCR array for fibrosis genes in AMVFs. AMVFs were treated with (**A**) ±10 μM compound 147 or (**B**) ±siRNA to ATF6; all cultures were treated for 48 h, then analyzed by a qRT-PCR array as described in the Methods (Section 4). In (**A**,**B**) green and red dots represent up- and downregulated genes, respectively. (**C**) A subset of differentially regulated genes from panels A and B; green and red numbers represent the fold up- or downregulation, respectively.

**Figure 8 ijms-21-01373-f008:**
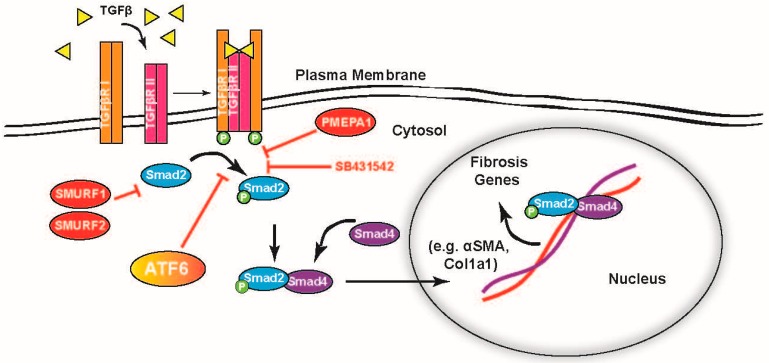
Representation of ATF6 acting on the TGFβ–Smad pathway for induction of fibrosis genes. TGFβ is a ligand for TGFβ receptor II, which upon binding forms a receptor complex with TGFβ receptor I, activating their kinase function. The TGFβ receptor complex phosphorylates receptor-Smads 2 and 3, which then complex with co-Smad 4 and move to the nucleus where they activate fibrosis genes (black arrows). These data suggest ATF6 activity inhibits the phosphorylation of the receptor-Smads at or before the level of the TGFβ chemical inhibitor SB431542 (red “T” arrows).

**Table 1 ijms-21-01373-t001:** Forward and reverse primers used for qRT-PCR, listed 5′ to 3′.

Primer	Sequence	Primer	Sequence
*Acta2*—Fwd	5′-GTTCAGTGGTGCCTCTGTCA-3′	*Pmepa1*—Fwd	5′-TGTCCTCGGAAGGATGCCTCTGG-3′
*Acta2*—Rev	5′-ACTGGGACGACATGGAAAAG-3′	*Pmepa1*—Rev	5′-CAGCGAGTCGGTCAGTGGGC-3′
*Atf6*—Fwd	5′-CTTCCTCCAGTTGCTCCATC-3′	*Smurf1*—Fwd	5′-AGGCTCTGCAAGGCTCTACAG-3′
*Atf6*—Rev	5′-CAACTCCTCAGGAACGTGCT-3′	*Smurf1*—Rev	5′-GGTGGTTGTGAGCAAGACTCTG-3′
*Col1a1*—Fwd	5′-AAGACGGGACGGCGAGTGCT-3′	*Smurf2*—Fwd	5′-AAGAACTACACAGTGGGAACGC-3′
*Col1a1*—Rev	5′-TCTCACCGGGCAGACCTCGG-3′	*Smurf2*—Rev	5′-CACGTTGCACCATTTGTTCC-3′
*Gapdh*—Fwd	5′-ATGTTCCAGTATGACTCCACT-3′	*Tnnt2*—Fwd	5′-GGAAGAGACAGACAGAGAGAGA-3′
*Gapdh*—Rev	5′-GAAGACACCAGTAGACTCCAC-3′	*Tnnt2*—Rev	5′-GGTTTCGCAGAACGTTGATTT-3′
*Tcf21*—Fwd	5′-CATTCACCCAGTCAACCTGA-3′		
*Tcf21*—Rev	5′-CCACTTCCTTCAGGTCATTCTC-3′

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
