# Peer review of "The ER Unfolded Protein Response Effector, ATF6, Reduces Cardiac Fibrosis and Decreases Activation of Cardiac Fibroblasts"

_ijms, 2020, doi:10.3390/ijms21041373_

Round 1

Reviewer 1 Report

The manuscript by Stauffer et al addresses the role of ER stress response protein ATF6 in cardiac fibrosis. The authors showed previously the protective role of ATF6 in cardiac myocytes. In this manuscript they address specific role of ATF6 in cardiac fibroblasts. The paper is well-written and can be of interest for readers. However, there is a number of technical issues that must be addressed.

1. There is no reference for Ethics approval of mouse experiments.

2. Figure legends must be more informative. Generally through the manuscript, the time of the treatment is not given neither in the text, nor in figure legends or Methods.

3. Figure 2. Quantification must be evaluated statistically and presented with the significance level. The number of animals per group is not indicated. There are no scale bars. How can you explain low fibrosis in myocyte-specific ATF6-KO (Fig. 2B)?

4. Figure 3. The time of the treatment is not indicated. Changes of the expression should be verified also on protein level by western blotting or other technique.

5. Why was the Collagen disk contraction assay performed with NIH3T3 cells and not with fibroblasts isolated from KO mice? It would be more direct evidence. It is stated in the Methods that disks were evaluated at “12-hours intervals”. What is the time point shown in the Figure 4A?

6. Figure 4D – Quantification must be evaluated statistically. Time of the treatment is not given.

7. Figure 5. Figure legend is not clear. In case the “graphs depict the quantification of immunoblots”, the immunoblot for ATF6 is missing.

8. Please, indicate in the Figures, which specific pairs of means were compared using Newman-Keuls test.

Reviewer 2 Report

The manuscript id: IJMS-690337, titled “The ER Unfolded protein response Effector, ATF6, reduces cardiac fibrosis and moderates activation of cardiac fibroblasts” by Stauffer et al., has demonstrated the ATF6 inhibitory effects on TGFB1/pSMAD2 signaling.

There is no consistency in Col-1a expression compared to other profibrotic genes The authors may have to test pSMAD3 levels. Figure 1 the legends of the bar charts had not consistency. PSR staining in the two WT mice (Fig 1A and B) are showing two different directions. Figure 4: Please include scale bars and arrow arrow heads to show cell contraction Is there a way to show ATF6 activation after compound 147 treatment? The isolated fibroblast may be tested for their purity Figure 9 should be modified with no pSMAD3 There are typos should be fixed

Round 2

Reviewer 1 Report

The authors addressed some of the raised issues. However, the data shown in Fig. 2 were obtained from a single animal and therefore cannot be assessed statistically. Given the variability usually present in mouse models, this is not acceptable.

It is still not indicated how many cells/viewpoints have been assessed for quantifications of immunocytochemistry images in Fig. 3.
